# Haploidentical HSCT in the Treatment of Pediatric Hematological Disorders

**DOI:** 10.3390/ijms25126380

**Published:** 2024-06-09

**Authors:** Anna Marszołek, Maria Leśniak, Anna Sekunda, Aleksander Siwek, Zuzanna Skiba, Monika Lejman, Joanna Zawitkowska

**Affiliations:** 1Student Scientific Society of Department of Pediatric Hematology, Oncology and Transplantology, Medical University of Lublin, 20-093 Lublin, Poland; anna.marszolek@gmail.com (A.M.); marialesniak68@gmail.com (M.L.); annasekunda00@gmail.com (A.S.); aleksiwek12@gmail.com (A.S.); zuzannaskiba01@o2.pl (Z.S.); 2Independent Laboratory of Genetic Diagnostics, Medical University of Lublin, 20-093 Lublin, Poland; monika.lejman@umlub.pl; 3Department of Pediatric Hematology, Oncology and Transplantology, Medical University of Lublin, 20-093 Lublin, Poland

**Keywords:** haplo-HSCT, GVHD, pediatrics, pediatric hematology, leukemia, AML, transplantation

## Abstract

Allogeneic hematopoietic stem cell transplantation has become a treatment option for otherwise non-curative conditions, both malignant and benign, affecting children and adults. Nevertheless, the latest research has been focusing extensively on transplantation from related and unrelated haploidentical donors, suitable for patients requiring emergent hematopoietic stem cell transplantation (HSCT) in the absence of an HLA-matched donor. Haploidentical HSCT (haplo-HSCT) can be an effective treatment for non-malignant pediatric disorders, such as primary immunodeficiencies or hemoglobinopathies, by enabling a much quicker selection of the appropriate donor for virtually all patients, low incidence of graft-versus-host disease (GVHD), and transplant-related mortality (TRM). Moreover, the outcomes of haplo-HSCT among children with hematological malignancies have improved radically. The most demanding tasks for clinicians are minimizing T-cell-mediated alloreactivity as well as early GVHD prevention. As a result, several T-cell depletion approaches, such as ex vivo T-cell depletion (TCD), and T-cell replete approaches, such as a combination of anti-thymocyte globulin (ATG), post-transplantation cyclophosphamide (PTCy), cyclosporine/tacrolimus, mycophenolate mofetil, or methotrexate, have been taken up. As more research is needed to establish the most beneficial form of therapy, haplo-HSCT is currently considered an alternative donor strategy for pediatric and adult patients with complications like viral and bacterial infections, invasive fungal disease, and GVHD.

## 1. Introduction

Hematopoietic stem cell transplantation (HSCT) became a treatment option over 60 years ago, and due to immense progress in the medical field, it has been used to treat various diseases, including both malignant and non-malignant hematological conditions [1]. The procedure is based on the administration of hematopoietic stem cells from the donor to replace the recipient’s malfunctioning hematopoietic system [2]. HSCT can be categorized depending on the relationship between the donor and recipient, as well as the source of the graft [3]. The great limitation in the general application of HSCT is the difficulty in finding a compatible donor in the context of the human leukocyte antigen (HLA) [4]. The concept of haploidentical HSCT (haplo-HSCT) was developed to serve as an alternative to allogenic HSCT (allo-HSCT), requiring an HLA-identical donor, whereas haplo-HSCT is conducted with at least a half-matched identical donor [5]. Histocompatibility genes are inherited as a group (haplotype) both from the mother and the father (Figure 1) [6,7].

Minimal residual disease (MRD), defined as the lasting presence of leukemia cells after a chemotherapy/radiotherapy regimen, can be used to evaluate the risk of relapse and reaction to a therapy plan [8]. The assessment of the MRD status before transplantation is a crucial and significant prognostic factor in children and adolescents with hematological malignancies, as patients with negative MRD before haplo-HSCT tend to achieve substantially longer overall survival (OS) and event-free survival (EFS) [9].

The greatest advantage of haplo-HSCT is that it considerably widens the pool of potential donors, as over 95% of patients can find a donor among their parents, siblings, or children who share one HLA haplotype with the patient. Such donors are easily accessible, which is particularly beneficial for patients who are in an urgent need of HSCT [10]. Nevertheless, due to imperfectly matched HLA, haplo-HSCT is more prone to graft-versus-host disease (GVHD) and poorer graft function [5]. The negative response of the patient’s immune system is prompted by alloreactive T-cells, either from the donor against the recipient’s tissues or from the patient against the graft [11]. High tumor-related mortality (TMR) has primarily been associated with acute GVHD (aGVHD), whereas chronic GVHD (cGVHD) is said to be the leading cause of poor quality of life of children who have received haplo-HSCT [12].

## 2. GVHD Management

Although haplo-HSCT significantly increases the number of possible donors and offers a low toxicity rate and stable engraftment, it involves a greater risk of GVHD and treatment-related fatality. Children are affected by transplant-related toxicities at a very young age, which has life-long consequences, with cGVHD being one of the most serious complications [13]. Previously, it was believed that total T lymphocyte depletion from the graft was the only method of GVHD prophylaxis. Despite being successful, such an approach led to an increased rate of severe infections and patient mortality [14,15].

To provide higher rates of successful engraftment and better GVHD management, different strategies regarding alloreactive T-cells were developed, resulting in haplo-HSCT no longer being perceived as a last-resort treatment [16]. Presently, there are two main pathways of T-cell depletion (TCD), ex vivo and in vivo.

The ex vivo pathway involves ${CD34}^{+}$ cell selection with a subsequent infusion of a megadose of purified ${CD34}^{+}$ cells, ${CD3/CD19}^{-}$ negative selection, and T-cell receptor (TCR) α/β with CD19 depletion, as well as depletion of CD45RA naive T-cells [17,18,19,20]. In the past, ${CD34}^{+}$ cell selection was the most frequently used method. Even though it was associated with a lower risk of GVHD, it could delay the patient’s immune reconstitution and slow down viral clearance [21]. The latest study results acknowledge haplo-HSCT with TCRαβ and CD19 depletion as a choice for pediatric patients with non-malignant hematological and immunological conditions, due to satisfactory immune reconstitution [22]. This kind of graft manipulation results in stem cell transplants that are highly enriched in natural killer (NK) cells and ${TCR\gamma /\delta }^{+}$ T-cells that are essential for viral protection [23,24,25]. Additionally, the depletion of B lymphocytes with anti-CD19 is performed to decrease the risk of Epstein–Barr virus (EBV) reactivation [26].

A recent study indicates that GVHD, particularly after TCRα/β and CD19 depletion haplo-HSCT, can be managed by the administration of multiple doses of zoledronic acid in the post-transplant course. The substance is said to enhance ${TCR\gamma /\delta }^{+}$ lymphocyte action after depleted haplo-HSCT without life-threatening side effects, such as commonly observed asymptomatic hypocalcemia and flu-like symptoms after infusion. The research proved that three or more infusions of zoledronic acid were associated with a decreased frequency of aGVHD and cGVHD, as well as reduced transplant-related mortality (TRM), compared to one or two doses. As zoledronic acid is safe and provides a positive post-transplant response, it can be administered to children a few weeks after haplo-HSCT [27].

The in vivo pathway covers methods based on low-dose post-transplantation cyclophosphamide (PTCy) and anti-thymocyte globulin (ATG) treatment, as well as a combination of PTCy and ATG with cyclosporine (CsA) and mycophenolate mofetil (MMF) [18,19,20]. Recent research suggests that the PTCy-based method provides better leukemia-free survival (LFS) as well as GVHD-free survival [19]. The positive effect of haplo-HSCT with PTCy on GVHD risk has been proven in both adult and pediatric patients who qualified for a transplant [28]. Cyclophosphamide is also used in various non-malignant hematological disorders, not only in sickle cell disease (SCD), but also in thalassemia, primary immunodeficiencies (PIDs), severe aplastic anemia (SAA), and osteopetrosis [26,29]. The effect caused by PTCy is selective depletion of alloreactive donor T-cells that are responsible for GVHD and graft rejection, while preserving the non-alloreactive resting memory T-cells responsible for adaptive immunity and the blood stem cells necessary for successful engraftment. The crucial part of the process is the administration of cyclophosphamide during the appropriate time frame [25]. It is most often administered on the 3rd and 4th days after graft transplantation [30,31]. GVHD prophylaxis may also include tacrolimus and MMF administered during the 5th post-transplant day [30]. Furthermore, even though TCRα/β and CD19 depletion are also affiliated with successful GVHD prevention in children and adults, the application of PTCy enables haplo-HSCT without ex vivo TCD [13,32]. It has been observed that maternal- and collateral-related donor transplants lead to a greater prevalence of GVHD and reduced survival, in comparison to the direct family. Nonetheless, compared to other ATG-based or PTCy-based regimens, the low-dose PTCy/ATG regimen considerably constricted the cumulative incidence of aGVHD and did not worsen the relapse risk, thus making maternal/collateral donor transplants a safer option [33].

Another approach using basiliximab, a chimeric monoclonal antibody aiming at alpha chain of interleukin-2 receptors (CD25 antigen), has been proposed as a form of treatment of steroid-refractory (SR) aGVHD [34]. Basiliximab is used in both adult and pediatric patients after haplo-HSCT as a satisfactory second-line treatment. It has been demonstrated that the patient response rate to corticosteroids reaches only 50%, whereas SR aGVHD is responsible for a long-term mortality rate as high as 70%. As basiliximab mostly affects activated T lymphocytes without suppressing the resting T lymphocytes, it does not alter engraftment or the incidence of infectious complications. The effectiveness of SR aGVHD management differs depending on the distressed organ, with the best response to basiliximab being in skin involvement, followed by gut and liver involvement. Furthermore, no significant adverse effects were observed, thus enabling the usage of basiliximab as a recommended strategy [35].

## 3. Malignant Disorders

Despite the continuous development of new treatment options for pediatric acute lymphoblastic (ALL) and myeloid leukemia (AML), there is still a group of patients who are not eligible for aggressive induction chemotherapy, which results in a poor overall prognosis and a limited choice of therapies [36,37], and for whom HSCT remains a relevant and effective medical procedure [38,39]. Although allo-HSCT continues to be the first-line treatment, haplo-HSCT is gaining increased attention, due to its many advantages, such as a significant increase in the number of possible donors, low toxicity rate, wide availability for therapeutic procedures after transplantation, and a greater spectrum of immunologic reactions against tumor cells [40].

### 3.1. Haplo-HSCT with Ex Vivo T-Cell Depletion

The first approaches to performing bone marrow transplantation from related, not fully matched donors in patients with hematologic malignancies consisted of a conditioning regimen analogous to the one used with matched sibling donors; however, they did not yield satisfactory results, because of poor OS rates [41]. The implementation of transplantation TCD led to improved GVHD prevention in children with severe combined immunodeficiency (SCID) [21], but could not be used in patients with leukemia due to frequent graft failure, despite the benefits of improved GVHD prevention [42,43]. Eventually, a haplo-HSCT with the positive selection of ${CD34}^{+}$ peripheral stem cells preceded by myeloablative conditioning including myeloablative drugs, such as the alkylating agent thiotepa, total-body irradiation (TBI), busulfan-based conditioning regimens, and advanced engraftment in the treatment of adult and pediatric AML [44,45].

In recent years, the transplantation of T- and B-cell-depleted allografts from haploidentical family donors was assessed by Lang et al. in a prospective Phase II trial in pediatric patients with acute leukemias and myelodysplastic syndrome. A total of 46 patients received ${CD3}^{+}/{CD19}^{+}$-depleted peripheral allografts after a melphalan-based conditioning regimen. Twelve patients survived free of disease, with a median follow-up of 4.3 years; however, over 60% of patients died, with relapse being the most common cause of death. In terms of GVHD, a greater rate was observed in comparison with a historical cohort of patients who received CD34-selected grafts with similar numbers of stem cells (26% vs. 7% grades II–IV and 21% vs. 13% cGVHD) [46]. On the other hand, the incidence of GVHD was more favorable than in two similar pediatric trials regarding ${CD3}^{+}/{CD20}^{+}$-depleted allografts with serotherapy [47] and ${CD3}^{+}/{CD19}^{+}$-depleted allografts without serotherapy [48] (41% and 33% aGVHD II–IV and 22% and 13% cGVHD, respectively) [49].

Currently, novel studies aim to test a new graft manipulation involving ${ꭤ\beta }^{+}$ T-cell removal, while retaining both NK and ${\gamma \delta }^{+}$ T-cells in the graft [35]. During a Phase I–II clinical trial (NCT01810120), Locatelli et al. noted that patients who underwent TCR ꭤβ/CD19-depleted haplo-HSCT achieved survival outcomes comparable to patients who received an unmanipulated transplant from an HLA-compatible donor. In addition, this type of transplant was associated with a low incidence of aGVHD as well as cGVHD [50]. Seven years later, after a longer follow-up period and the use of the aforementioned transplant, the authors reanalyzed 213 pediatric patients diagnosed with ALL (n = 152) and AML (n = 61). The median age at the time of transplantation was 9.5 years. All patients were in morphologic complete remission (CR) at the time of transplantation. A total of 18 patients were positive for MRD before transplantation. All patients received fully conditioned myeloablation including a regimen based on TBI and/or cytostatic treatment. None of the patients received prophylactic pharmacotherapy for the prevention of GVHD, and aGVHD grades II–III occurred with a combined incidence of 14.7%. A total of 129 children developed viral infections due to cytomegalovirus (CMV), adenovirus (ADV), and human herpesvirus 6 (HHV-6). It is worth mentioning that the survival rates or relapse rates did not differ between patients who developed or did not develop the mentioned infections. The main reason for treatment failure was relapse. Children diagnosed with ALL who had low/negative MRD prior to HSCT, early-stage disease prior to transplantation, and received a TBI-based myeloablative conditioning regimen had a prolonged disease-free survival (DFS). In addition, it was noted that, compared to other regimens, the use of TBI, thiotepa, and fludarabine improved the DFS rate [51]. Another study regarding TCD graft modification compared outcomes of HLA-matched unrelated donor (MUD) transplantation and ex vivo ${ꭤ\beta }^{+}$ T-cell-depleted haplo-HSCT among pediatric patients diagnosed with high-risk acute leukemias. The 5-year OS and relapse-free survival (RFS) rates in the group of haplo-HSCT patients were similar to those in the MUD group. After transplantation, 15 of the 53 patients died, including 5 who received haplo-HSCT. The most common cause of death was relapse. Furthermore, the incidence rates of grades II–IV aGVHD and cGVHD occurrence in MUD patients were higher than in haplo-HSCT patients, while GVHD prophylaxis differed between the groups [27]. As of right now, there is no evidence as to which platform, ex vivo T-cell depletion or PTCy, is better. A retrospective and multicenter study by Pérez-Martínez et al. intended to compare the viability of two cohort groups of haplo-HSCT platforms in children and young adults with high-risk hematological malignancies and PTCy and ex vivo TCD grafts. The second cohort of patients was divided into four groups, each of which received differently modified T-cells: highly purified ${CD34}^{+}$ cells (n = 13), combined depletion of ${CD3}^{+}$ donor T-cells and ${CD19}^{+}$ B-cells (n = 82), combined depletion of ꭤβ T-cells and ${CD19}^{+}$ B-cells (n = 34), and highly purified ${CD34}^{+}$ cell transplantation with the addition of a product depleted of naive ${CD45RA}^{+}$ lymphocytes to the transplant. Only patients receiving haplo-HSCT in the form of repeated transplantation were given ATG as a preparative regimen. The other patients received total lymphoid irradiation (TLI), TBI, or high doses of methylprednisolone. The cumulative incidence rates of aGVHD grades I–II or grades III–IV were significantly lower in the case of children treated with TCD; however, the 2-year OS and DFS rates did not particularly differ between patient groups, as both major platforms for haplo-HSCT were proven to be effective. Based on the study, a young donor, negative MRD, and myeloid pathology are promising prognostic factors [52].

${ꭤ\beta }^{+}$ T-cell-depleted transplantation is expected to sustain anti-tumor and anti-infectious features, due to the ability of ${CD3}^{+}$ cells with TCR ${\gamma \delta }^{+}$ receptors to impose graft-versus-tumor activity [53,54,55]. Greater recovery of ${\gamma \delta }^{+}$ T-cells after transplantation has been connected with better EFS, fewer infectious complications, and longer DFS rates in both pediatric and adult populations [56,57]. Such findings correlate with the results of a study conducted by Dadi et al. to investigate the impact of increased ${\gamma \delta }^{+}$ T-cell content in the engraftment. Researchers analyzed the treatment outcomes in 38 children and adolescents diagnosed with acute leukemias who underwent haplo-HSCT with depletion of ${ꭤ\beta }^{+}$ T-cells and ${CD19}^{+}$ B-cells. Six patients in this study received a reduced-intensity conditioning regimen that included fludarabine, thiotepa, melphalan, and ATG. However, because of frequent graft failure, the TBI regimen with additional drugs was implemented. The study demonstrated that a graft with a content of ${\gamma \delta }^{+}$ T-cells equal or higher than 9.5 × 10^6^/kg was associated with notably higher EFS rates [58].

It is crucial to optimize treatment approaches for high-risk treatment failure patients to reduce the risk of relapse. Patients undergoing transplantation during an active stage of ALL have previously obtained highly unsatisfying results [46,59]. The reported survival rates after haplo-HSCT in relapsed or chemotherapy-resistant AML patients fluctuated between 9% and 44% [60,61]. Because ATG given at the time of the graft infusion may inactivate grafted NK and ${\gamma \delta }^{+}$ cells, replacing it with non-lymphodepleting targeted immunomodulation could protect the transplant. Promising results were achieved by Shelikhova et al., who analyzed 22 pediatric patients diagnosed with primary refractory (n = 10) or relapsed refractory (n = 12) AML in active disease status, who were administered haplo-HSCT with peripheral blood stem cells engineered by the depletion of ${ꭤ\beta }^{+}$ T-cells and ${CD19}^{+}$ B-cells. Patients received fludarabine and cytarabine as a preparative regimen followed by treosulfan and thiotepa as a myeloablative conditioning regimen. Instead of ATG, patients were given tocilizumab, and an additional 10 children received abatacept. The infusion of donor lymphocytes containing a CD45RA-depleted fraction with or without a hypomethylating agent was used as prophylactic therapy post-transplantation. A total of 21 patients (95%) achieved CR. At the two-year follow-up, rates such as TRM, EFS, and OS were 9%, 49%, and 53%, respectively. The cumulative incidence rates of grades II–IV aGVHD and cGVHD were 18% and 23% [62].

Hematopoietic grafts infused with regulatory T-cells (Tregs) have been discovered to be protected from GVHD without decreasing conventional T-cell (Tcon) antileukemic activity in both murine and preclinical studies [63,64,65]. Pierini et al. researched outcomes of haplo-HSCT infused with donor Tcons under the protection of previously transferred donor Tregs as the exclusive GVHD prophylaxis. As expected, a significantly low relapse rate and a 75% cGVHD/relapse-free survival (CRFS) rate in adult AML patients were noted [66]. In this regard, Massei MS et al. aimed to test the efficacy of a newly developed haplo-HSCT variant with adoptive immunotherapy with thymic-derived ${CD4}^{+}$ ${CD25}^{+}$ ${FoxP3}^{+}$ Treg and Tcons in 20 children and adolescents diagnosed with high-risk ALL and AML. The conditioning regimen was based on TBI and chemotherapy. A total of 25% of the patients developed ≥grade II aGVHD, including one patient who developed grades III/IV and died 35 days after transplantation. A total of 75% of the patients included in the described analysis are alive and leukemia-free with a median follow-up time of 2.1 years. The feasibility of CRFS was 79% [67] (Table 1).

### 3.2. Haplo-HSCT without Ex Vivo T-Cell Depletion

Due to the often-delayed reconstitution of the immune system, resulting in infectious complications and graft failure following TCD haplo-HSCT, current research is focusing extensively on developing new conditioning regimens and intensified GVHD prophylaxis combined with the T-cell replete approach (Table 2).

#### 3.2.1. Granulocyte Colony-Stimulating Factor (G-CSF)/ATG-Based Protocol

Huang et al. proposed a novel strategy of GVHD prevention by incorporating G-CSF and ATG with unmanipulated allografts in a pilot cohort study known as the “Beijing Protocol” [68]. All patients achieved sustained, full donor-type engraftment. The authors noted 37.9% and 5.2% aGVHD rates (grades II–IV and III–IV) and a 61.9% cGVHD rate [69]. Studies have shown that a G-CSF (filgrastim) addition amplifies monocytic phenotype myeloid-derived suppressor cell (M-MDSC), promyelocytic-MDSC (P-MDSC), and granulocytic-MDSC (G-MDSC) expansion [70,71]. G-CSF is said to activate a subset of ${CD34}^{+}$ cells with monocyte functions [72]; moreover, it causes bone marrow regulatory B-cells (Bregs) to produce higher levels of interleukin 10 (IL-10) and transforming growth factor beta (TGF-β) [73]. Bregs influence T-cell tolerance by inhibiting T-cells of the Th1 phenotype in favor of the Th2 phenotype and heightening Treg levels [74]. All of the above-mentioned mechanisms of the G-CSF effects on the grafts were proven to lower the incidence of aGVHD and cGVHD [70,71,72,73,74]. Because the treatment approach has become successful, considerable progress has since been made to elongate patient survival [75,76,77]. Regarding AML, a study by Liu et al. declared a 73.3% 5-year OS rate in pediatric patients who underwent T-cell replete haplo-HSCT [75]. To focus more on a high-risk group of patients, a multicenter retrospective study was taken up to compare the outcomes between children who were exposed to unmanipulated haplo-HSCT with a G-CSF/ATG-based regimen or an identical sibling donor (ISD)-HSCT. Both transplant methods displayed comparable OS and DFS rates [78]. Bai et al. analyzed 200 pediatric patients diagnosed with high-risk AML who underwent their first unmanipulated haplo-HSCT. A total of 103 patients had a nondetectable MRD status prior to HSCT. The 4-year OS, EFS, and cumulative incidence of relapse (CIR) rates were 71,9%, 62,3%, and 32,4%, respectively. The above rates differed significantly between the two groups of patients, depending on the MRD status before transplantation. Patients who were MRD-negative achieved better OS, EFS, and CIR rates of 80.5%, 73.3%, and 23.8%, respectively, compared to MRD-positive children pre-transplant, for whom the aforementioned rates were 63.4%, 51.4%, and 41.0%, respectively [9].

BCR/ABL-positive ALL occurred in 3–5% of all pediatric ALL patients [79] and was associated with a poor overall outcome [80] before tyrosine kinase inhibitors (TKIs) came into play. Chen et al. performed haplo-HSCT in 50 Philadelphia chromosome-positive (${Ph}^{+}$) ALL children. Patients received two myeloablative conditioning regimens, TBI or a combination of cytarabine, busulfan, and semustine (Me-CCNU), both combined with ATG. The 3-year incidence of relapse and non-relapse mortality rates were 22.7% and 16.4% [81].

Another study was conducted in order to compare the treatment outcomes between haplo-HSCT- and TKI-based chemotherapy for ${Ph}^{+}$ pediatric ALL patients. Mild grades I–II aGVHD and severe grades III–IV aGVHD occurred in 25 and 5 out of 68 patients undergoing haplo-HSCT, respectively. A total of 42 patients survived more than 100 days after transplantation. The CIR, EFS, and OS rates were 23.5%, 73.4%, and 80.3% in the transplantation group. The authors found no significant difference in the rates of the above-mentioned indicators between patients belonging to the standard risk group, regardless of whether they had a transplant or not. However, in patients with at least one adverse prognostic factor, the above rates were better in those who underwent transplantation, which showed that only in the high-risk group of patients did haplo-HSCT have a significant advantage in terms of prolonged survival [82].

Studies test the efficacy of the G-CSF/ATG regimen during haplo-HSCT in various diseases, many of which with a negative prognosis. Once more, Xue et al. compared the results of unmanipulated haplo-HSCT with chemotherapy, this time among children with very high-risk (VHR) Philadelphia chromosome-negative (${Ph}^{-}$) B-ALL. The 3-year estimated OS and EFS rates were better in children in the haplo-HSCT group than in the intensive chemotherapy group at 80.6% vs. 62.4% and 81.0% vs. 52.0%, respectively [83]. In another study, Bai et al. analyzed the outcomes of 38 patients with B-ALL with t(v;11q23) MLL-rearrangements. Similarly, the 4-year estimated OS and EFS rates were significantly greater in the haplo-HSCT cohort than in the intensive chemotherapy cohort [84].

A study by Hu GH et al. advocated the safety of combining chimeric antigen receptor T-cell (CAR-T) therapy with haplo-HSCT in patients suffering from relapsed B-ALL [85], which is said to have an extremely poor prognosis and be resistant to chemotherapy [86]. Earlier studies linked the MRD status with post-transplantation outcomes; therefore, relapsed B-ALL patients with lesser or undetectable MRD obtained significantly higher EFS and lower CIR rates [87]. In treatment based on the MRD status, the authors analyzed the effect of haplo-HSCT after CAR-T therapy or chemotherapy on the long-term survival and safety of pediatric patients with the first relapse of B-ALL. A total of 40 children were included in the analysis, 26 of whom received CAR-T-cells. At the time of the first remission, negative minimal residual disease was obtained by 21 patients in the CAR-T group and 10 patients in the chemotherapy group. The median MRD before haplo-HSCT in the group receiving chemotherapy (n = 14) was significantly higher than in the CAR-T group (n = 26). Post-haplo-HSCT complications in the form of infection, treatment-related mortality, aGVHD, and cGVHD were not significantly different between the two patient groups. A significant advantage was observed in the increase of 3-year leukemia-free survival (LFS) and 3-year OS in patients who received CAR-T therapy before haplo-HSCT, compared to patients receiving chemotherapy: (71.8% vs. 44.4%) and (84.6% vs. 40%), respectively. The results of the aforementioned study may suggest that the administration of CAR-T for positive pre-HSCT MRD has a beneficial effect on patient survival [88].

*KMT2A* is a frequently rearranged gene in leukemias, mostly in pediatric and infant AML [89,90]. In 21 patients diagnosed with AML with *t(v;11q23)/KMT2A,* rearrangement transplantation options were evaluated. A total of 17 of the 21 children underwent haplo-HSCT transplantation, while the others had a matched sibling donor. The Beijing protocol was used as myeloablative treatment, with no deaths related to myeloablation. Fully matched HSCT appeared to have a lower death rate, but the authors found no significant differences in the OS, EFS, or CIR rates between patient cohorts. The severity of aGVHD was higher in the group undergoing haplo-HSCT; however, only one haploidentical transplant patient died from aGVHD-related complications [91].

Rare acute leukemias occurring in children prior to 3 years of age are called infant leukemias [92]. While myeloid leukemias have similar outcomes in most children, the ones deriving from lymphoid progenitors have poorer disease progression and relapse-free survival, in comparison with ALL in older patients [93,94]. Allo-HSCT remains the first-line treatment for relapsed ALL [95]. To determine the safety of haplo-HSCT in such patient groups, the authors retrospectively analyzed a group of 97 infants and patients under 3 years of age diagnosed with acute leukemia. The most frequently identified gene fusion was *KMT2A* rearrangement found in 37 patients, of which 15 were infants. In children without the aforementioned gene fusion, the 3-year DFS and OS rates were significantly lower than in children with the identified *KMT2A* rearrangement at 63.8% vs. 78.4% and 66.9% vs. 86.1%, respectively. The median follow-up was 45 months. The 3-year OS and DFS rates in children diagnosed at <1 year of age and children diagnosed at ≥1 year were almost the same at 82.5% vs. 72.8% and 77.8% vs. 66.3%, respectively [96].

#### 3.2.2. PTCy-Based Protocol

Cyclophosphamide (Cy) has been extensively studied for its immunosuppressive characteristics [97] in the context of bone marrow transplantation, and it is said to lower the chances of graft rejection, as well as GVHD, if given at an optimal time [98,99].

The first study to introduce the PTCy-based protocol was a clinical trial carried out by O’Donnell et al. regarding high-risk hematologic malignancy patients undergoing haplo-HSCT. Conditioning included fludarabine followed by TBI, PTCy that was given as GVHD prophylaxis at day 3 at a dose of 50 mg/kg with Mesna (80% of Cy dose in four divided doses over 8 h), mycophenolate mofetil, and tacrolimus. A total of 8 out of 13 patients obtained sustained engraftment [100]. Based on data from adult studies, haplo-HSCT with PTCy significantly reduces the risk of cGVHD and appears to be a better option than a MUD [101,102]. Sachdev et al. retrospectively examined the outcomes of 15 pediatric patients diagnosed with high-risk/relapsed ALL (n = 10) and AML (n = 5). A total of 3 patients received MUD transplantation, and 12 underwent haplo-HSCT. All patients accepted the transplant, except one who died before the procedure due to pneumonia. A total of 11 children achieved remission and were alive at a median follow-up time of 775 days. The cumulative incidences of aGVHD and cGVHD were 57.1% and 21.4%, respectively. The rates of relapse or mortality did not depend on the type of conditioning regimen used. The OS and EFS rates were 80% and 73.3% [103]. In another study, comparing haplo-HSCT and chemotherapy in children diagnosed with intermediate risk (IR) AML, the patients in the chemotherapy group had a higher CIR and worse EFS, but roughly equivalent OS. The results of the study indicated that haplo-HSCT is a beneficial therapeutic option for children with IR-AML in first complete remission (CR1), particularly for those patients who have MRD ≥ 10^−3^ after induction therapy [104]. Another study compared the treatment outcomes in 80 pediatric patients diagnosed with high-risk acute leukemia who, after myeloablative conditioning based on busulfan, received a transplant with PTCy from either not fully matched-related or MUD. The grades II–IV and grades III–IV aGVHD rates in the MUD group were higher than in the haploidentical related donor (HRD) group, i.e., 48.9% vs. 34.3% and 8.9% vs. 2.9%, respectively. The CIR of overall cGVHD and moderate to severe cGVHD was almost the same in both groups. Because of the similar outcomes and survival rates in both groups, PTCy haplo-HSCT should be given to patients requiring bone marrow transplantation who do not have an HLA-matched related or unrelated donor [28]. Tannumsaeung et al. aimed to determine the results of haplo-HSCT with PTCy preceded by either TBI- or thiotepa-based conditioning regimens. The percentage of graft-related complications and infections was comparable in both groups of patients [105].

### 3.3. Other Studies on Haplo-HSCT in Hematologic Malignancies

Apart from the above-mentioned trials or retrospective analyses, the available literature discussing haplo-HSCT in pediatric patients with hematologic malignancies contains a couple of single-institution reports (Table 3).

**Table 2 ijms-25-06380-t002:** Novel studies on haplo-HSCT without ex vivo T-cell depletion.

Number of Patients	Median Age	Transplant Type	Conditioning Regimen and GVHD Prophylaxis	Incidence of aGVHD	Incidence of cGVHD	Relapse NRM	OS Rate	Reference
179	ISD-HSCT 11 haplo-HSCT 12	ISD-HSCT (n = 23) haplo-HSCT (n = 156)	Ara-C + Bu + Cy + Me-CCNU + G-CSF + ATG + CSA + MMF + short-term MTX	ISD-HSCT II–IV aGVHD 13% haplo-HSCT II–IV aGVHD 34.8%	ISD-HSCT 14.1% haplo-HSCT 34.9%	Relapse ISD-HSCT 39.1% haplo-HSCT 16.4% NRM ISD-HSCT 0% haplo-HSCT 10.6%	3-year OS ISD-HSCT 73% haplo-HSCT 74.6%	[78]
200	10	haplo-HSCT	Ara-C + Bu + Cy + Me-CCNU + ATG + CSA + MMF + short-term MTX + G-CSF	II–IV aGVHD 41.1% III–IV aGVHD 9.5%	56.1%	Relapse 31% NRM 5%	4-year OS 80.5%	[9]
68	10	HSCT: haplo-HSCT (n = 37) MSD (n = 3) UCB (n = 4) chemotherapy and TKi (n = 24)	Ara-C + Bu + Cy + Me-CCNU + CSA + MMF + hydroxyurea (MSD) + ATG (UCB, haplo-HSCT) + G-CSF (MUD, haplo-HSCT) + short-term MTX (MSD, haplo-HSCT) + MP (UCB)	I–II aGVHD 56.8% III–IV aGVHD 11.4%	40.9%	Relapse no transplant: 45.8% transplant: 13.5%	3-year OS 80.3%	[82]
104	7	haplo-HSCT (n = 42) chemotherapy (n = 62)	Ara-C + Bu + Cy + Me-CCNU + ATG + G-CSF + CSA + MMF + short-term MTX	II–IV aGVHD 54.8% III–IV aGVHD 11.7%	55.8%	Relapse haplo-HSCT: 11.9% chemotherapy: 51.6% NRM haplo-HSCT: 9.5% chemotherapy: 0%	predicted 3-year OS haplo-HSCT: 80.6% chemotherapy: 62.4%	[83]
38	4	haplo-HSCT (n = 18) MUDT (n = 1) Chemotherapy (n = 18)	Ara-C + Bu + Cy + Me-CCNU + ATG + CSA + MMF + short-term MTX	II–IV aGVHD 37%	53.8%	predicted 4-year CIR 39.1%	predicted 4-year OS 69.8%	[84]
40	CAR-T-cell therapy before haplo-HSCT 9.5 chemotherapy before haplo-HSCT 9.0	CAR-T-cell therapy before haplo-HSCT (n = 26) chemotherapy before haplo-HSCT (n = 14)	TBI or Ara-C + Bu + Cy + ATG + Me-CCNU + G-CSF + CSA + MMF + short-term MTX	CAR-T-cells: II–IV aGVHD 26% chemotherapy: II–IV aGVHD 23%	CAR-T-cells: 53% chemotherapy: 50%	Relapse CAR-T-cells: 26.9% chemotherapy: 50% NRM CAR-T-cells: 3.8% chemotherapy: 7.14%	3-year OS CAR-T-cells: 84.6% chemotherapy: 40%	[88]
21	4.4	haplo-HSCT (n = 17) MSDT (n = 4)	Ara-C + Bu + Cy + Me-CCNU + ATG, CSA + MMF + short-term MTX (n = 17) Ara-C + Bu + Cy + CSA + short-term MTX (n = 4)	haplo-HSCT II–IV aGVHD 76.4%	No information	Relapse 14.3% NRM 9.5%	100%	[91]
97	Patients diagnosed <1 year: 1.4 Patients diagnosed ≥1 year: 2.2	haplo-HSCT	Ara-C + Bu + Cy + Me-CCNU + ATG + CSA + MMF + short-term MTX	II–IV aGVHD 45.3% III–IV aGVHD 8.8%	36%	NRM 3.1% 3-year CIR 26.9%	3-year OS 74.2%	[96]
15	8.5	MUD (n = 3) haplo-HSCT (n = 12)	TBI + Cy, Flu + Bu, Flu + Bu + L-PAM, TT + Flu + Cy + TBI, Flu + Cy + TBI, PTCy + CSA/TAC + MMF, MTX + CSA + ATG + G-SCF,	57.1%	21.4%	NRM 26.7%	80%	[103]
80	haplo-HSCT: 10 chemotherapy: 8	haplo-HSCT (n = 33) chemotherapy (n = 47)	Ara-C + Bu + Cy + Me-CCNU + ATG	II–IV aGVHD 44.6% III–IV aGVHD 6%	60.6%	3-year CIR 25.4%	3-year OS 85.4%	[104]
80	haplo-HSCT with PTCy 7 MUD 8.9	haplo-HSCT with PTCy (n = 35) MUD (n = 45)	Bu + Flu ± etoposide + ATG + TAC + MMF, Bu + Flu + Cy + PTCy + TAC + MTX	MUD: II–IV aGVHD 48.9% III–IV aGVHD 8.9% haplo-HSCT with PTCy: II–IV aGVHD 34.3% III–IV aGVHD 2.9%	MUD: 18.3% haplo-HSCT with PTCy: 11.4%	Relapse MUD: 28% haplo-HSCT with PTCy: 25.6% NRM MUD: 2,2% haplo-HSCT with PTCy: 0%	3-year OS MUD: 83.7% haplo-HSCT with PTCy: 88.6%	[28]
43	TBI regimen 8.7 TT regimen 8.8	haplo-HSCT with PTCy	TBI + Cy + Flu + L-PAM, TT + Flu + Bu, PTCy + MMF + calcineurin inhibitor or sirolimus + G-CSF	aGVHD TBI regimen: 56.5% TT regimen: 45%	TBI regimen: 21.7% TT regimen: 10%	NRM TBI regimen 17.4% TT regimen 20%	3-year OS 62.4%	[105]

aGVHD—acute graft-versus-host disease, Ara-C—cytarabine, ATG—anti-thymocyte globulin, Bu—busulfan, CAR-T—chimeric antigen receptor T-cells, cGVHD—chronic graft-versus-host disease, CIR—cumulative incidence of relapse/progression, CSA—cyclosporin A, Cy—cyclophosphamide, Flu—fludarabine, G-CSF—granulocyte colony-stimulating factor, haplo-HSCT—haploidentical hematopoietic stem cell transplantation, ISD-HSCT—identical sibling donor hematopoietic stem cell transplantation, L-PAM—melphalan, Me-CCNU—methyl chloride hexamethylene urea nitrate (semustine), MMF—mycophenolate mofetil, MP—methylprednisolone, MUD—matched unrelated donor, MSDT—matched sibling donor transplant, MTX—methotrexate, NRM—non-relapse mortality, OS—overall survival, PTCy—post-transplant cyclophosphamide, TAC—tacrolimus, TBI—total body irradiation, TKi—tyrosine kinase inhibitor, TT—thiotepa, UCB—unrelated cord blood.

**Table 3 ijms-25-06380-t003:** Small cohort studies reporting haplo-HSCT results in the treatment of hematologic malignancies in children.

Condition/Disease	Study Group	Graft Characteristics	Transplant Type	Conditioning Regimen and GVHD Prophylaxis	Survival Rate	References
MDS: RCC, refractory anemia with excess of blasts	8 patients, 7 F, 1 M median age: 6.4 years	MNCs, × 10⁸/kg—19.8 The median ${CD34}^{+}$ cells, 10⁶/kg—11.8	haplo-HSCT with PTCy	Cy at +3 and +4 days after transplantation, low-dose TAC, MMF	OS = 100% DFS = 100%	[106]
MDS: RCC, advanced MDS, MDR-AML	27 patients, 15 F, 12 M median age: 10 years	The median MNCs, ×10⁸/kg—8.6 The median ${CD34}^{+}$ cells, 10⁶/kg—2.9	haplo-HSCT	Ara-C + Bu + Cy + Me-CCNU + ATG + CSA + MMF + short-term MTX	estimated 3-year rate probabilities for 10 patients diagnosed with RCC OS = 90% DFS = 90%, for 17 patients with advanced MDS/MDR-AML OS = 77.7% DFS = 77.7%	[107]
HLH	Case report First patient: 8 months/male Second patient: 10 years/male	The dose of ${CD34}^{+}$ cells, 10⁶/kg First patient: 24.33 Second patient: 11.96	haplo-HSCT with PTCy	Cy at +3 and +4 days after transplantation + MMF, First patient: CSA + Flu + Treo + alemtuzumab Second patient: TAC + RTX + ATG + Flu + Cy + TBI	disease-free period of 912 and 239 days for the First and Second patients	[108]
HLH	12 patients 7 F, 5 M median age: 4.5 years	No information	haplo-HSCT with ${ꭤ\beta }^{+}$/${CD19}^{+}$ cell depletion (n = 2) and PTCy (n = 10)	Cy at +3 and +4 days after transplantation + Flu + Treo + ATG + TT (${ꭤ\beta }^{+}$/${CD19}^{+}$ cell depletion group) + TBI (PTCy group)	OS = 66.7%	[109]

Ara-C—cytarabine, ATG—anti-thymocyte globulin, CSA—cyclosporine A, Cy—cyclophosphamide, F—female, Flu—fludarabine, HLH—hemophagocytic lymphohistiocytosis, M—male, MDR-AML—myelodysplasia-related acute myeloid leukemia, MDS—myelodysplatic syndrome, Me-CCNU—methyl chloride hexamethylene urea nitrate (semustine), MMF—mycophenolate mofetil, MNCs—median mononuclear cells, MTX—methotrexate, PTCy—post-transplant cyclophosphamide, RCC—refractory cytopenia of childhood, RTX—rituximab, TAC—tacrolimus, TBI—total body irradiation, Treo—treosulfan, TT—thiotepa.

## 4. Non-Malignant Disorders 

Haplo-HSCT may be an effective treatment option not only for malignancies. Its wider application in pediatric non-malignant disorders has been thoroughly described in the literature. Haplo-HSCT usage ranges from primary immunodeficiencies (PIDs) to all kinds of anemias.

### 4.1. Primary Immunodeficiencies

PIDs comprise a large, heterogeneous group of diseases affecting all components of the immune system that may be caused by changes in over 300 genes [110,111]. One treatment option for the patients is hematopoietic stem cell transplantation [112]. The best option is a graft from a healthy HLA-genoidentical matched sibling donor. However, such transplants are possible in no more than 25% of the cases. Furthermore, less than 70% of the remaining patients will have a suitable matched unrelated donor, and chances are even slimmer for patients belonging to certain ethnic groups [113]. To avoid the delay due to a prolonged search for a matching donor, an HSCT from an HLA-haploidentical family donor (HIFD) can be performed, despite the fact that such a procedure is associated with a higher risk of GVHD and graft failure, which may lead to increased mortality rates [114,115,116]. The available literature discussing haploidentical HSCT in pediatric patients with PID consists of single-institution reports and multicenter retrospective analyses [117,118,119,120,121,122] (Table 4).

In the reviewed studies, researchers analyzed the impact of HSCT on patients with PIDs, ranging from trials where children were diagnosed with 12 different types of PIDs [117] to ones including only SCID patients [120]. The survival rates of patients after PID treatment with HSCT from an HLA-haploidentical donor ranged from 84% to 62.7%. There were numerous factors that contributed to the causes of death among patients, i.e., CMV infection, progressive respiratory failure, GVHD, or thrombotic microangiopathy. Regarding infectious complications after transplantation, one study reported a 63.6% incidence rate of post-HSCT CMV viremia [118], while another reported a 58.8% cumulative incidence rate of CMV and ADV infections [117]. At the same time, the incidence rate of significant GVHD in almost all the studies remained low. The rates of GVHD ranged from 54% to 27.2%; however, most of the studies revealed rates of GVHD lower than 40%. On top of that, most patients who developed a GVHD experienced it at a mild I or II grade. The time needed for immune reconstitution was also satisfactory [117,118,119,120,121,122]. One study reported that 75% (six out of eight) of survivors had a whole-blood chimerism greater than 95% [118], and in another trial, at the last follow-up, 76.1% of patients had a full donor chimerism [117]. However, in the case of an adenosine deaminase (ADA) deficiency, a type of PID, the results of haploidentical HSCT treatment were inconclusive. One study presented the OS rate of 68.4% (13/19 patients) [120], whereas the trial conducted by Hassan et al. suggested that haplo-HSCT is not an effective form of treatment in this PID type. The results of 106 patients showed that the HSCT OS rates from matched sibling and family donors were 86% and 81%, respectively. This proved to be a better result in comparison with patients receiving HSCT from haploidentical donors, many of who presented graft failure, with an OS rate of 43%. Having said that, long-term immune recovery showed that, regardless of the transplant type, the overall T-cell numbers were similar. Moreover, humoral immunity and donor B-cell engraftment were achieved in nearly all the survivors [123]. Those data stand in contrast to the satisfactory results of haploidentical HSCT in other types of immunodeficiencies. An explanation can include the fact that in the trial conducted on mice, the microenvironment of ADA-deficient bone marrow showed a reduced capacity to support in vitro or in vivo hematopoiesis. Therefore, patients with an ADA deficiency may fail to support engraftment of the transplanted HSCT [124].

In conclusion, a ${CD3}^{+}{TCR\alpha \beta^{+}/CD19}^{+}$-depleted HSCT from an HLA-haploidentical donor is an effective form of treatment for children with PIDs, especially when finding a matched sibling donor is not possible.

### 4.2. Sickle Cell Disease

The course of SCD is highly variable. However, the one feature that unites the diverse nature of this disease is a shortened life expectancy and significant morbidity. Patients with SCD may experience hemolytic anemia, severe pain, vaso-occlusive crises, stroke, avascular necrosis, pulmonary hypertension, infections, renal failure, and thrombosis. Despite many genetic attempts, HSCT remains the only curative method [125]. The best outcomes are observed after HLA-matched sibling transplantation. However, approximately 70% of patients with SCD do not have an HLA-matched sibling. Alternative donor options include matched unrelated donor, unrelated umbilical cord blood transplantation, and mismatched related transplantation [14,126].

According to studies, haplo-HSCT exhibits high efficiency among pediatric patients. In the clinical trial carried out by J. Foell et al., 15 out of the 20 studied patients with high-risk SCD received a CD3/CD19-depleted graft and the rest received a TCRαβ/CD19-depleted graft. All patients experienced primary engraftments with stable grafts. The median time to reach leukocyte recovery was 16 days, and neutrophile recovery occurred after 19 days, and thrombocyte recovery occurred 10 days after the transplant. When it comes to lymphocytes, the median time of ${CD3}^{+}$ T-cell recovery was 173 days post-transplantation; it was 277 days for ${CD4}^{+}$ T-cells and 213 days for ${CD8}^{+}$ T-cells. B lymphocytes were restored around the 80th day after transplantation. According to these data, leucocytes, neutrophils, and thrombocytes recover rapidly, while T-cell recovery appears delayed. In this study group, the OS, EFS, and DFS were 90%. Total respiratory morbidity (TRM) was observed in 10% of the patients [127]. In another study conducted with a 9-year follow-up, haplo-HSCT did not bring satisfactory outcomes. The trial enrolled 22 patients with symptomatic SCD, 8 out of whom were recipients of a parental haploidentical donor graft. The median age of the patients was 9.0 (±5.0) years. All patients achieved donor engraftment with a 100% donor chimerism, with a median time to full engraftment amounting to 12.5 days. A total of 50% of the patients, however, developed evidence of graft rejection on approximately the 30th day post-transplant and required an additional stem cell infusion. After the infusion, 25% of the patients recovered with a sustained 100% donor chimerism, whereas the other patients progressed to graft failure with recurrence of disease. Donor engraftment was eventually sustained in 62% of the patients, and graft failure occurred in 38% of the subjects. A follow-up visit after 9 years post-transplantation indicated that 75% of the patients were alive, with 37.5% eventually experiencing sustained engraftment and remaining disease-free. However, SCD recurrence took place in another 37.5% of the studied population. The OS rate amounted to 75% [128].

Pawlowska et al. reported a complete engraftment with a 99.9% to 100% donor chimerism after T-cell replete haploidentical stem cell infusion. All patients maintained stable engraftment at the last follow-up (11 months). Additionally, neutrophil engraftment occurred between the 14th and 26th days post-transplant. Half of the patients presented high levels of donor-specific anti-HLA antibodies, which required the implementation of an antibody management protocol. This enabled neutrophil engraftment on days 16 and 26, respectively [30]. Similar results were reached by another research group [129]. Their follow-up was carried out 11, 14, and 30 months after HSCT transplant. The survival rate amounted to 100%, and all of the patients exhibited a complete donor chimerism. What is important, they did not have any more symptoms of SCD. Neutrophil engraftment was noted on days 12, 17, and 20. However, both of the studies above consisted of a low number of patients; therefore, the reliability of the results cannot be fully confirmed.

Despite the introduction of prophylaxis, GVHD still remains a critical issue. Nonetheless, it does not occur as often as previously and with significantly lower intensity. In a study of three patients, mild skin GVHD was observed in one patient. None of those patients experienced cGVHD, nor central nervous system toxicity. Surprisingly, two patients experienced an asymptomatic CMV reactivation [129]. In a study of four patients, one developed aGVHD grade I, while three of the patients experienced mild skin GVHD and responded well to immunosuppression therapy. Moreover, in three of the patients, HHV-6 was detected, but resolved spontaneously without any treatment [30]. In the far-reaching study of eight patients, among which five underwent a successful engraftment, four experienced aGVHD. Two of those patients developed grade I, while the other two developed grade II aGVHD. Subsequently, three of those four patients developed cGVHD and two of them died from complications. In the study comparing T-haplo-HSCT and matched sibling donor grafts, 35% of the patients in the T-haplo-HSCT group experienced aGVHD. However, in all cases, GVHD resolved after a few days of low-dose prednisolone applications. None of the patients developed aGVHD grades III–IV. Nonetheless, four developed a steroid-sensitive cGVHD with cutaneous, oral, ocular, and fascial involvement responding to primary treatment. Apart from GVHD, haplo-HSCT complications included CMV, BK virus (BKV), EBV, HHV6 and ADV reactivation, BKV-associated nephritis, rotavirus diarrhea, CMV pneumonitis, and macrophage activation syndrome with a late graft failure [127].

These data indicate that cyclophosphamide, tacrolimus, and MMF usage is effective and extremely beneficial for patients with SCD as prophylaxis. Although it does not always guarantee 100% efficiency, such a treatment contributes to the development of less severe GVHD.

### 4.3. Severe Aplastic Anemia

Severe aplastic anemia (SAA) is a state of bone marrow failure resulting from its hypoplasia or aplasia. As a consequence, the bone marrow does not produce enough erythrocytes, leucocytes, and/or thrombocytes. Recently, SAA outcomes have significantly improved because of HSCT introduction and immunosuppressive treatments. HLA-matched sibling donor HSCT is a primary treatment option for pediatric SAA patients. However, due to the frequent lack of such donors, haplo-HSCT is most commonly performed instead [130]. Based on the little data available, this treatment seems to be very successful. In the multicenter trial conducted among 35 patients, all of them reached full engraftment with the median time for myeloid recovery amounting to 14 days and for platelet recovery to 18 days [131]. Another study proved a 98% success rate; however, 6% of the patients had a secondary graft rejection after more than 20 days post-transplant [130].

While haplo-HSCT usage decreases the severity of SAA symptoms, it also increases the risk of GVHD or post-transplant hemophagocytic syndrome (PTHPS), despite cyclophosphamide prophylaxis [131,132,133]. In both studies, the GVHD rate amounted to more than 50% of the cases. It included not only aGVHD grades I to IV, but also chronic GVHD involving the skin, liver, and gastrointestinal tract [130,131]. Nonetheless, previous research proved that the introduction of an immunosuppressive drug, inhibitor of mammalian target of rapamycin (mTOR), such as sirolimus on the 8th day before the transplant, significantly reduces this complication among adult patients [132,133]. Regarding pediatric patients, a co-stimulation blockade with cytotoxic T lymphocyte antigen-4 (CTLA4) IgA is a promising method of reducing GVHD and PTHPS occurrence. This approach has been described, inter alia, by Jaiswal et al. In their trial, two groups were compared. The first group received an extended T-cell co-stimulation blockade (COSBL) with abatacept, sirolimus, and post-transplantation cyclophosphamide, while the other—the control group—only received post-transplant cyclophosphamide. Both groups included 10 patients with SAA. The patients treated within the COSBL protocol had much better outcomes than those in the control group in terms of GVHD occurrence. The incidences of aGVHD were 10.5% and 50%, respectively. Moreover, approximately 12.5% of the patients in the COSBL group experienced cGVHD, while in the control group, it was 56% of the patients [132]. Those results confirm the previous thesis. Haplo-HSCT in combination with therapy involving COSBL, abatacept, sirolimus, and cyclophosphamide significantly decreases the GVHD incidence in pediatric patients with SAA.

Apart from GVHD, many patients experienced different complications of haplo-HSCT, such as bacterial, fungal, and viral (CMV and EBV) infections [131,133]. Wang et al. additionally described a reversible posterior leukoencephalopathy syndrome that occurred in two patients in that study and ended with their death [131].

When it comes to the OS rate of haplo-HSCT used in SAA, the values fluctuated from 70% to approximately 86% [130,131,132].

### 4.4. Fanconi Anemia

Fanconi anemia (FA) is a rare, inherited condition characterized by congenital malformations, progressive marrow failure, and a predisposition to solid tumors, as well as acute myelogenous leukemia. Patients with FA are excellent candidates for haplo-HSCT because, without such a treatment, the prognosis is extremely poor [134,135]. The study of Ayas et al. enrolled 19 pediatric patients with FA. All of the patients experienced neutrophil engraftment at the median time of 14 days. On the other hand, platelet recovery occurred in 18 patients at the median time of 20.5 days. The overall incidence of aGVHD amounted to 42%, while extensive cGVHD occurred in only one patient. The follow-up examination was performed at the median time of 38 months, and the OS rate was set at 89% [134]. Another study included 24 pediatric patients who underwent haplo-HSCT for FA. Full engraftment was achieved in 22 patients (92%), with the median time for neutrophil and platelet recovery of 12 and 10 days, respectively. Eventual primary graft failure occurred in two patients. Four patients (16%) developed aGVHD grades I and II. Additionally, of the 22 patients at risk, only 1 developed mild, skin-only cGVHD (4.5%). On the follow-up after a median time of 5 years, all patients were alive; thus, the OS rate was 100%. Eleven patients (45.8%) developed post-HSCT viral reactivations or infections [136].

On the basis of this trial, a connection was made among older age, high transfusion burden, previous androgen exposure, development of clonal evolution, and lower survival rates. Moreover, children with FA undergoing haplo-HSCT are associated with a higher risk of developing severe aGVHD, due to the underlying DNA repair defect and deregulation of the apoptotic process [136].

## 5. Future Directions

Although haplo-HSCT became a point of interest over 20 years ago, to this day, there are several aspects that require further investigation, particularly in pediatric patients [137]. A lot of research focuses on the comparison of transplant results among MUD, HLA mismatched unrelated donor (MMUD), and HRD to determine the best choice for a patient lacking a family HLA-matched donor. For pediatric patients with acute leukemia, the 5-year OS rates between T-replete haplo-HSCT and transplant from MUD and MMUD appeared comparable. The relapse incidence (RI) and NRM also showed similar results [138]. Another study revealed that the results of haplo-HSCT with PTCy and the MAC regimen and the MUD HSCT with the ATG regimen were similar. Haplo-HSCT is recognized as an advantageous option for pediatric patients with high-risk leukemia, lacking a related or an unrelated HLA-identical donor. Moreover, a comparison between PTCy-based haplo-HSCT with MAC and non-MAC regimens exhibited similar toxicity [28]. CGVHD was observed more often in patients undergoing an MUD transplant rather than after haplo-HSCT. Pediatric patients with hematological malignancies can be successfully treated with haplo-HSCT, as the aGVHD and cGVHD incidence rates and OS rates are satisfactory [139].

Research on post-transplant immune recovery is still developing, as it is crucial to decrease the prevalence of infectious complications and relapse risks. Innate immunity, especially including NK cells, that exert both anti-leukemic and anti-viral effects, is responsible for such recovery. Despite that, TCD methods for haplo-HSCT cause around a 2-month delay for NK cells to be produced from donor hematopoietic stem cells. The study was conducted to determine if lower doses of anti-T lymphocyte globulin (ALTG) would affect the NK cell count after transplantation. It was revealed that the number of NK cells was indeed substantially elevated, which led to the conclusion that a higher concentration of ALTG might cause a lengthened immune readjustment. Nevertheless, because ALTG has a favorable effect on GVHD and graft rejection, it would be optimal to incorporate high pre-transplant ALTG doses with low post-transplant ALTG doses [140].

Another aspect worth further investigating is haplo-HSCT followed by donor lymphocyte infusion (DLI), which presents a distinct graft-versus-leukemia (GVL) effect. Data gathered from adult patients showed that ATG-based haplo-HSCT with subsequent prophylactic DLI reduced the risk of relapse, in addition to raising long-term survival among high-risk acute leukemia patients without enhancing the toxicity of the therapy [141]. Pediatric data are rather limited; however, a few studies were conducted and the results were analogous to those for the adult patients. Prophylactic DLI provided successful and safe outcomes in children with both malignant and non-malignant hematological diseases [142,143].

Moreover, haplo-HSCT is recognized as an independent risk factor for CMV infection. Statistically, the frequency of infections is higher after haplo-HSCT than after HLA-matched HSCT, which contributes to the extended mortality rate among pediatric patients [144]. Furthermore, the usage of PTCy is also associated with a greater rate of CMV infection, which suggested the need for an active prophylaxis regimen. As CMV-active anti-viral drugs, like valganciclovir or ganciclovir, present severe side effects, adding to the toxicity of the treatment, a new drug was developed. CMV DNA terminase inhibitor, letermovir, was introduced as prophylactic medication for HSCT recipients, including PTCy-based haplo-HSCT. The research showed that letermovir decreased the occurrence of CMV-related complications, as well as the necessity to use CMV-specific treatment, without altering the TRM or OS [145,146]. Data regarding the use of letermovir in pediatric patients are insufficient. However, recent studies confirm that it can be profitably and safely administered to children as a preventative measure against CMV reactivations [147,148].

## 6. Conclusions

Due to many its advantages, haplo-HSCT is becoming an increasingly popular form of transplantation in the absence of quick access to fully compatible donors. Although further research in the form of multicenter studies is needed to assemble larger, homogenous research groups with standardized peri-transplant treatment regimens to identify optimal GVHD prophylaxis and conditioning strategies, haplo-HSCT has proven to be an effective form of bone marrow transplantation in hematologic malignancies, both in active and refractory diseases with the OS and RFS rates similar to the MUD grafts. Moreover, it has been applied successfully in various non-malignant hematological disorders, such as SCD, PIDs, and SAA. Current research is focusing on post-transplant immune recovery to reduce the risk of infectious complications and relapses.

## Figures and Tables

**Figure 1 ijms-25-06380-f001:**
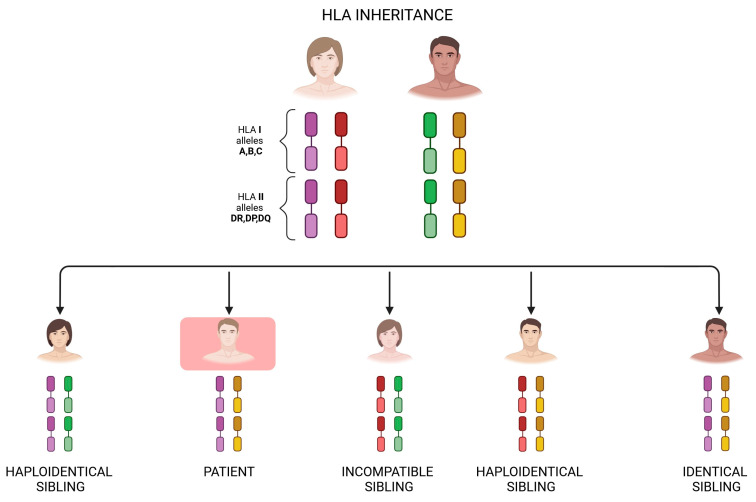
HLA gene inheritance. Image created with biorender.com (accessed on 2 April 2024). HLA genes are inherited according to classic Mendelian genetics, with HLA molecules being codominantly expressed. An HLA-haploidentical donor is a related donor who shares exactly 1 HLA haplotype and differs by a variable number of HLA genes on the unshared haplotype. Siblings have a 25% chance of being identical at the genotypic level of HLA molecules.

**Table 1 ijms-25-06380-t001:** Novel studies on haplo-HSCT with ex vivo T-cell depletion.

Number of Patients	Median Age	Transplant Type	Conditioning Regimen and GVHD Prophylaxis	Incidence of aGVHD	Incidence of cGVHD	Relapse NRM	OS Rate	Reference
80	9.7	haplo-HSCT with TCD	TBI + TT + Flu, TBI + TT + L-PAM, TT + Bu + Flu, Bu + Cy + L-PAM	I–II aGVHD 30%	5%	relapse 24% NRM 5%	Predicted 5-year OS 72%	[50]
213	9.5	haplo-HSCT with ${ꭤ\beta }^{+}$/${CD19}^{+}$ cell depletion	TBI + TT + Flu, TBI + TT + L-PAM, Bu + TT + Flu, Bu + Cy + L-PAM	II–III aGVHD 14.7%	8.1%	5-year NRM 5.2% 5-year CIR 22.7%	predicted 10-year OS 75.4%	[51]
53	No information	MUD (n = 32) haplo-HSCT with ${ꭤ\beta }^{+}$ cell depletion (n = 21)	Bu + Cy ± r-ATG, Flu + TT + L-PAM + ATG + RTX	MUD: II–IV aGVHD 53.1% haplo-HSCT: II–IV GVHD 33.3%	MUD: 25% haplo-HSCT: 19%	Relapse MUD: 15.6% haplo-HSCT: 9.5% NRM MUD: 21.9% haplo-HSCT: 14.3%	MUD: 68.8% haplo-HSCT: 76.2%	[27]
192	8.6	haplo-HSCT with PTCy (n = 41) haplo-HSCT with TCD (n = 151)	Flu + TT + L-PAM + Bu + high-dose PT-Cy + MMF + TAC + mesna, Flu + TT + L-PAM + Bu + TLI or TBI or high-dose MP or ATG	PTCy: I–II aGVHD 52.6% III–IV aGVHD 28.2% TCD: I–II aGVHD 30.6% III–IV aGVHD 14.7%	2-year cGVHD PTCy: 47.7% TCD: 28.6%	predicted 2-year relapse PTCy: 26.8% TCD: 31.1%	2-year OS PTCy: 65.4% TCD: 52%	[52]
38	8.5	haplo-HSCT with ${ꭤ\beta }^{+}$/${CD19}^{+}$ cell depletion	Flu + TT + L-PAM + ATG + RTX, Flu + TT + Treo + ATG + RTX ± TBI	45%	7%	5-year NRM 36%	5-year OS 51%	[58]
22	9.6	PBSC haplo-HSCT with ${ꭤ\beta }^{+}$/${CD19}^{+}$ cell depletion	Flu + Ara-C + Treo + L-PAM + TT + RTX + bortezomib + tocilizumab + abatacept	II–IV aGVHD 18%	23%	2-year CIR 42%	2-year OS 53%	[62]
20	14.5	haplo-HSCT with adoptive immunotherapy with thymic-derived ${CD4}^{+}$${CD25}^{+}{FoxP3}^{+}$ Tregs and Tcons	TBI + TT + Flu + Cy + RTX	≥II aGVHD 25%	5%	relapse 5% NRM 15%	No information	[67]

aGVHD—acute graft-versus-host disease, Ara-C—cytarabine, ATG—anti-thymocyte globulin, Bu—busulfan, cGVHD—chronic graft-versus-host disease, CIR—cumulative incidence of relapse/progression, Cy—cyclophosphamide, Flu—fludarabine, haplo-HSCT—haploidentical hematopoietic stem cell transplantation, L-PAM—melphalan, MMF—mycophenolate mofetil, MP—methylprednisolone, NRM—non-relapse mortality, OS—overall survival, PBSC—peripheral blood stem cells, PTCy—post-transplant cyclophosphamide, RTX—rituximab, TAC—tacrolimus, TBI—total body irradiation, Tcons—conventional T-cells, TLI—total lymphoid irradiation, Tregs—regulatory T-cells, Treo—treosulfan, TT—thiotepa.

**Table 4 ijms-25-06380-t004:** Data summary regarding overall survival, graft modification, number of patients with GVHD prophylaxis, type of conditioning, cell dose, incidence and type of GVHD, and mean time of B- and T-cell recovery in different trials. Ranked by increasing survival rates.

Trial Name	Number of Patients Receiving Haplo-HSCT	Type of Graft Modification	Number of Patients with GVHD Prophylaxis	Type of Conditioning (Number of Patients Receiving at Least One of Listed Drug)	Cell Dose (Mean; Rage Dose)	Overall Survival Rate (%) (Number of Patients)	Incidence of GVHD (%) (Number of Patients)	Most Common Grade of GVHD (%)	Mean Time of T-Cell Recovery (days)	Mean Time of B-Cell Recovery (days)
Shah et al. [117]	25	${TCR\alpha \beta^{+}/CD19}^{+}$	22	Treo, Flu, TT, ATG, Alemtuzumab 23	CD3: 3.3 (0.075–9.5) × 10^4^/kg CD34: 17.8(4.7–50.9) × 10^6^/kg	84 21/25	47.8 11/23	II	129	85
Brettig et al. [118]	11	${TCR\alpha \beta^{+}/CD19}^{+}$	No information	Treo, Flu, TT 11	CD3: 1.5 (0.3–2) × 10^4^/kg CD34: 14.1 (3.7–20.8) × 10^6^/kg	81.8 8/11	27.2 3/11	II	132	93
Buckley et al. [119]	77	No information	0	0	No information	78 60/77	36.3 28/77	I or II	90–120	No information
Buckley et al. [120]	149	No information	0	0	No information	73 109/149	30.2 45/149	I or II	90–120	No information
Neven et at. [121]	22	No information	22	Flu, Bu, Cy, RTX, Alemtuzumab 22	CD3: 10.7 (2.6–30.2) × 10^7^/kg CD34: 13.8 (2.6–47.7) × 10^6^/kg	72.7 16/22	54 12/22	II	157	204
Holzer et al. [122]	19	CD34 selection; TCRαβ/CD19 depletion; TCRαβ depletion (+in vivo CD20-depletion); CD3/CD19 depletion	9	No information	No information	62.7 12/19	21 (acute) 15 (chronic)	I or II	30–365	No information

ATG—anti-thymocyte globulin, Bu—busulfan, Cy—cyclophosphamide, Flu—fludarabine, RTX—rituximab, Treo—treosulfan, TT—thiotepa.

## Data Availability

Not applicable.

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
