# Peer review of "Haploidentical HSCT in the Treatment of Pediatric Hematological Disorders"

_ijms, 2024, doi:10.3390/ijms25126380_

Round 1
Reviewer 1 Report
Comments and Suggestions for Authors
The manuscript " Haploidentical HSCT in the treatment of pediatric hematological disorders" reviews all the diverse types of haplo-HSCT transplants and their varied ways used in the clinic to avoid relapse post-transplant and graft versus host disease (GVHD) among children.
The authors showed an extensive list of references that covered the newest information from the field, gathering the information required to cover the treatments for patients undergoing HSCT.
The review is scientifically sound, and well-written, and the tables presented are easy to understand and help to gather the numerous data in the manuscript.
I have no major concerns.
Comments on the Quality of English Language
The English written is sound and I have not noticed any issues, however, is always best practice to review before the final version of the manuscript.
Reviewer 2 Report
Comments and Suggestions for Authors
This review article summarized the strengths and weaknesses of haploidentical HSCT in pediatric patients. The authors start with the GVHD (one of the main concerns of haplo-HSCT) and review the treatment outcomes according to the primary disease of the patients (malignancy or non-malignancies) and TCD methods. They provided a well-organized summary encompassing a wide range of topics related to haplo-HSCT in children.
However, I found quite a lot of typos and errors, and some data in the later tables were not sufficient. I got the impression that the article is still underprepared. please review the article again and check the errors and supplement insufficient data.
line 54, typo Minimal residual disease (MDR), MRD
line 76, need correction. total lymphocyte T depletion-> total T lymphocyte depletion
line 95, typo GVDH -> GVHD
table 1, at median age, typo, 9,7-> 9.7, 9,5-> 9.5
typo, relapse, NRM (%) 5,2-> 5.2, 22,7-> 22.7
reference 52, compared haploHCT with PTCy and TCD, but there is no comparative data
line 301, what is the 'graft relapse'?
line 335, it needs more detailed information of haploHSCT the patients received including what conditioning regimen and GVHD prophylaxis were given
line 388-397, the authors did not introduce the therapeutic outcome after haploHCT in infants with MLL rearrangement.
table 2, some conditioning regimens do not contain PTCy, what regimen did they use?
The summary of conditioning regimen is inconsistent.
table 3 minimal information regarding haplo-HSCT should be provided, at least of conditioning regimen, cell dose, GVHD prophylaxis, TCD methods.
line 457, what does the PID stand for? Please clarify the abbreviations.
table 4, more detailed information of haplo-HSCT needs to be summarized in the table.
Round 2
Reviewer 2 Report
Comments and Suggestions for Authors
Most of the comments were addressed.